# A Cross-Sectional Observational Study of the Relationship between Outdoor Exposure and Myopia in University Students, Measured by Conjunctival Ultraviolet Autofluorescence (CUVAF)

**DOI:** 10.3390/jcm11154264

**Published:** 2022-07-22

**Authors:** Valentina Bilbao-Malavé, Jorge González-Zamora, Elsa Gándara, Miriam de la Puente, Elena Escriche, Jaione Bezunartea, Ainara Marizkurrena, Elena Alonso, María Hernández, Patricia Fernández-Robredo, Manuel Sáenz de Viteri, Jesús Barrio-Barrio, Alfredo García-Layana, Sergio Recalde

**Affiliations:** 1Department of Opthalmology, Clínica Universidad de Navarra, 31008 Pamplona, Spain; vbilbao@unav.es (V.B.-M.); jgzamora@unav.es (J.G.-Z.); egandararod@unav.es (E.G.); mdelapuentec@unav.es (M.d.l.P.); jbezunartea@unav.es (J.B.); amsagaseta@unav.es (A.M.); ealonso2@unav.es (E.A.); mahersan@unav.es (M.H.); msaenzdevit@unav.es (M.S.d.V.); jbarrio@unav.es (J.B.-B.); aglayana@unav.es (A.G.-L.); srecalde@unav.es (S.R.); 2Retinal Pathologies and New Therapies Group, Experimental Ophthalmology Laboratory, Department of Ophthalmology, Universidad de Navarra, 31008 Pamplona, Spain; 3Faculty of Medicine, Universidad de Navarra, 31008 Pamplona, Spain; eescriche@alumni.unav.es; 4Navarra Institute for Health Research, IdiSNA, 31008 Pamplona, Spain; 5Red Temática de Investigación Cooperativa en Salud (RD16/0008/0011), Ministerio de Ciencia, Innovación y Universidades, Instituto de Salud Carlos III, 28029 Madrid, Spain

**Keywords:** conjunctival ultraviolet autofluorescence, high myopia, myopia, genetics, environmental factors, outdoor activities

## Abstract

Myopia is the most common refractive error worldwide. This cannot be explained by genetic factors alone, therefore, environmental factors may play an important role. Hence, the main objective of this study was to analyse whether outdoor exposure could exert a protective effect against the development of myopia in a cohort of young adults and to investigate ultraviolet autofluorescence (CUVAF), as a biomarker of time spent outdoors. A cross-sectional observational study was carried out using two cohorts. A total of 208 participants were recruited, 156 medical students and 52 environmental science students. The data showed that 66.66% of the medical students were myopic, while 50% of the environmental science students were myopic (*p* = 0.021). Environmental science students spent significantly more hours per week doing outdoor activities than medical students (*p* < 0.0001), but there was no significant difference with respect to near work activities between them. In both cohorts, the degree of myopia was inversely associated with CUVAF, and a statistically significant positive correlation was observed between spherical equivalent and CUVAF (Pearson’s r = 0.248). In conclusion, outdoor activities could reduce the onset and progression of myopia not only in children, but also in young adults. In addition, CUVAF represents an objective, non-invasive biomarker of outdoor exposure that is inversely associated with myopia.

## 1. Introduction

Myopia is the most common ocular disorder worldwide, and it is increasing alarmingly in prevalence, and reaching epidemic levels in many countries [1], not only in east Asian populations, but also in North America and Europe, particularly among young adults [2]. According to the World Health Organisation (WHO), by 2050, myopia and high myopia (HM) will affect 52% and 10%, respectively, of the world’s population [3]. This is concerning because myopic individuals have an increased risk of developing retinal detachment, glaucoma, cataracts, and structural complications such as myopic maculopathy, which is associated with severe visual impairment [4,5].

The exact mechanism underlying the development of myopia is not fully understood; it is thought that both genetic predisposition and environmental factors may play important roles. However, the recent increase in its prevalence, especially among young adults, may be primarily related to lifestyle changes, which include a combination of decreased time spent outdoors and increased near work activities [6]. Recent studies have found that reading at very close range for prolonged periods is associated with myopia [7,8,9]. However, to date, near work has been shown to make only a small contribution to the overall prevalence of myopia [10] and it has been demonstrated that children who performed higher amounts of near work, but also spend many hours doing outdoor activities, were still protected from the development of myopia [11].

Time spent outdoors is a well-studied environmental factor regarding the development and progression of myopia. Results from experimental animal models and epidemiological studies have suggested that the higher intensity light levels found outdoors may be the key factor, presumably through stimulation of retinal dopamine synthesis and release, which plays a role in the regulation of ocular growth [10]. Time spent outdoors has been shown to reduce the prevalence of childhood myopia [10,12,13,14,15]; however, it is not known whether spending more time outdoors during late adolescence and early adulthood could reduce the risk of myopia progression or the risk of late-onset myopia. This is a relevant time period as a significant proportion of myopia has been found to develop during adolescence and early adulthood [16,17] and very little research has been done on this topic [18,19,20].

To calculate time spent outdoors, most studies acquired their data through retrospective self-reported questionnaires, a potential source of recall bias and error [21,22]. Therefore, objective methods such as ultraviolet autofluorescence (CUVAF) have been recently investigated to quantify time spent outdoors [23,24,25]. Previous studies conducted in Australia and Norfolk Island showed that CUVAF size was inversely correlated with myopia [26,27]; however, these studies analysed a heterogeneous population ranging widely in age. Moreover, sun exposure depends on the geographic location, and to date, research in the Northern Hemisphere is limited [24,28,29].

Data from a survey at the Universidad de Navarra revealed a high prevalence of myopia and myopia progression during the university years among their students. These results make it of great importance to analyse what factors could be influencing this increase in the prevalence of myopia. Therefore, the main objective of this study was to analyse whether outdoor exposure could exert a protective effect against the development of myopia and HM in a population of young adults, specifically in university students, who usually spend a great amount of time doing near work. Moreover, CUVAF, as an objective measure of time spent outdoors, was also investigated in this population.

## 2. Materials and Methods

### 2.1. Study Design and Ethics Approval

A cross-sectional observational study was carried out using two different cohorts of students, one consisting of medical students and the other of environmental science students. The project was approved by the Institutional Review Board and the Ethics Committee of Clínica Universidad de Navarra (study code 2020.023) and the Faculties of Medicine and Science of the Universidad de Navarra. All procedures carried out conformed to the guidelines of the Declaration of Helsinki. All subjects were fully informed of the purpose and procedures of the study, and written informed consent was obtained for all participants.

### 2.2. Inclusion and Exclusion Criteria

All medical and environmental science students in the last 3 years of their degree programme at the Universidad de Navarra were invited to participate in the study from January 2020 to June 2021. The inclusion criteria were being a medical or environmental science student and having European ancestry. Exclusion criteria included secondary myopia or HM, anisometropia greater than 2.0 dioptres (D), and the presence of pterygium, pinguecula, previous conjunctiva surgery, or any conjunctival pigmented lesion that might make the measurement of the CUVAF area difficult.

### 2.3. Ophthalmic Exploration

All participants underwent an automatic objective refraction (Autorefractor Keratometer TRK-2P. Topcon Corporation, Tokyo, Japan), axial length (AL) measurement (IOLMaster; Carl Zeiss Meditec, Jena, Germany), and were asked to complete a questionnaire about their family history of myopia, wearing of spectacles, increase in myopia dioptres during their university years (subjective value provided by each participant), time spent doing near work and outdoor activities during a regular week, and sun exposure habits.

All subjects were classified according to their non-cycloplegic autorefraction as myopic (mean spherical equivalent refraction ≤−1.00 D [30]) or controls (mean spherical equivalent refraction > −1.00 D). This myopic threshold was selected to increase the probability to include only “truly myopic” subjects from a cohort of young students who still have a tendency to accommodate in the absence of cycloplegia. Myopic subjects were also classified according to their degree of myopia and AL as follows: low myopia (M1: −1.00 D to −3.00 D), moderate myopia (M2: −3.25 D to −5.75 D), and high myopia (HM ≤ −6.00 D or AL >26 mm).

### 2.4. CUVAF Acquisition and Measurement

The CUVAF area was measured in all participants using the BAF module on the Heidelberg Spectralis HRA + OCT (Heidelberg Engineering, Heidelberg, Baden-Württemberg, Germany), using the image acquisition protocol recently validated by Lingham et al. [30]. Quantification of the CUVAF area was carried out using a home-made plugin developed for Fiji/ImageJ 1.6v (NIH, Bethesda, MD, USA), an open-source Java-based image processing software [31]. CUVAF area measurement was performed by three different evaluators and the intra and interobserver reliability between them was assessed.

### 2.5. Genotyping

Genomic DNA was extracted from oral swabs using QIAcube (Qiagen, Hilden, Germany) and processed in the Ophthalmology Experimental Laboratory of the Clínica Universidad de Navarra. The SNP rs1060043, previously associated with CUVAF in Australian population [32], was genotyped by an ABI Prism 7300 Real-Time PCR System (Life Technologies, Carlsbad, CA, USA) using validated TaqMan assay C_9601723_30 (Applied Biosystems, Foster City, CA, USA).

### 2.6. Statistical Analyses

The general characteristics of the participants were compared using Student’s *t*-test and one-criterion ANOVA for continuous variables and Fisher’s F test for categorical variables. Pearson correlation tests were performed for the variables CUVAF area, refractive error, and AL. The frequencies of alleles and genotypes were calculated in all the groups and were compared using the chi-square test and Fisher’s exact test, and corresponding odds ratios (ORs) were calculated. The SNP analysed in this study was in Hardy–Weinberg equilibrium. For all statistical analyses, corrected *p* values < 0.05 (two-tailed) were considered statistically significant. Statistical analysis was performed using SPSS 20.1 Software (SPSS Inc., Chicago, IL, USA) and GraphPad Prism software version 5.0 (GraphPad Software Inc., San Diego, CA, USA).

A paired *t*-test was used to determine if there were differences between right and left eyes and nasal and temporal areas in terms of AL, refractive error, and mean CUVAF area. As no significant differences were found in the statistical analyses, we used the mean of both eyes as the reference value.

## 3. Results

### 3.1. Demographic Characteristics

A total of 228 participants were recruited. After eliminating those who did not meet the inclusion/exclusion criteria (19 non-European subjects and 1 that had conjunctival melanosis), 208 individuals were selected to enter the study, 156 were medical students and 52 were environmental science students. The data showed that 66.66% of the medical students were myopic, while 50% of the environmental science students were myopic (Table 1 and Table 2), with this difference being statistically significant (*p* = 0.021; Odds Ratio (OR); 2.03 (95% Confident Interval (CI) 1.1–3.6).

The myopic subjects in both groups of students were classified in M1, M2, and HM, which consisted of 46 (29.49%), 35 (22.44%), and 23 (14.74%) participants, respectively, in the medical students group, and 14 (26.92%), 6 (11.54%), and 6 (11.54%) in the environmental science students group (Table 1 and Table 2). When comparing the number of subjects in each myopic group between medical and environmental science students, the number of participants in M2 was significantly greater in the medical students cohort (*p* = 0.037; OR 95 = 0.41 (95% CI 0.2–0.9), while no statistically significant differences were found in M1 and HM groups between both cohorts of students.

The mean age of the medical and environmental science students was 22.38 ± 0.85 and 22.36 ± 1.73 years, respectively, and 69.87% and 69.23% of participants were female (Table 1 and Table 2), with no statistically significant differences between them in either case.

### 3.2. Environmental Factors

On average, medical students spent 57.61 ± 22.19 h per week doing near work activities and 9.03 ± 6.40 h per week doing outdoor activities. There was no significant difference in the time spent doing near work activities between myopic and controls (Figure 1A), but with respect to outdoor activities, individuals from the HM group spent significantly fewer hours per week outdoors than controls (*p* < 0.05) (Figure 1B). On the other hand, environmental students spent 55.30 ± 21.83 h per week doing near work activities and 17.15 ± 14.34 h per week doing outdoor activities. There was no significant difference in the time spent doing near work or outdoor activities between myopic and controls, although they follow the same pattern as medical students with myopic subjects reporting less time doing outdoor activities than controls (Figure 1C,D).

Comparing environmental factors between the cohorts of medical and environmental science students, we found that both myopic and control subjects in the environmental science students cohort spent significantly more hours per week doing outdoor activities in comparison to myopic and control subjects in the medical students cohort, respectively (*p* < 0.05 and *p* < 0.001) (Figure 2B). With respect to near work activities, there was no significant difference between myopic and control subjects when comparing the cohorts of medical and environmental science students (Figure 2A).

### 3.3. CUVAF

The mean CUVAF area of the entire study population was 2.93 ± 3.06 mm^2^ (range 0–9.64). In the cohort of medical students, the mean CUVAF area was 2.86 ± 3.23 mm^2^ (range 0–8.76), while the CUVAF area of the myopic and control groups were 2.33 ± 2.12 and 3.66 ± 4.41 mm^2^, respectively (Table 1). The mean CUVAF area of the myopic group was significantly smaller than that of the control group (*p* < 0.05) (Figure 3A). Furthermore, if the myopic are separated from the HMs, the latter had significantly smaller CUVAF areas than the controls (*p* < 0.05) (Figure 3B). This pattern is repeated when comparing the control group with the different myopic groups, where the higher the myopia, the smaller the CUVAF areas, and in the case of the HM group, this difference was statistically significant (*p* < 0.05) (Figure 3C).

In the cohort of environmental science students, the mean CUVAF area was 3.33 ± 2.64 mm^2^ (range 0–9.64), while the CUVAF area of the myopic and control groups were 2.64 ± 2.6 mm^2^ and 4.03 ± 2.59 mm^2^, respectively (Table 2). In a similar way to the medical students cohort, statistically significant differences were found in the CUVAF area between the control group and the different myopic groups (*p* < 0.05) following the same tendency of significantly smaller CUVAF areas in myopic and HM subjects in comparison with the control group (Figure 3D–F).

Comparing both student cohorts, environmental science students showed higher CUVAF areas (2.86 ± 3.23 mm^2^ in the medical students vs. 3.3 ± 2.64 mm^2^ in the environmental science students) but with no statistically significant difference, even when comparing control and myopic groups separately between both cohorts of students (Table 1 and Table 2).

The percentile analysis including all participants showed that the percentage of individuals with CUVAF 0 was significantly higher in the HM group compared to the control group (*p* < 0.001; OR = 14.17 (95% CI 1.54–129.6)). In addition, the frequency of individuals in the HM group who were below the 25th (*p* = 0.016; OR 95 = 4.2 (95% CI 1.4–12.9)) and 75th (*p* = 0.023; OR = 5.5 (95% CI 1.17–26.4)) percentile of CUVAF area was significantly higher compared to the control group (Figure 4).

A statistically significant positive correlation was observed between spherical equivalent and CUVAF area in both cohorts, so that the lower the spherical equivalent, the smaller the CUVAF area (Pearson’s r = 0.248 (95% CI 0.094–0.390); *p* = 0.001), while no clear correlation was found between CUVAF area and AL, or time spent outdoors (Figure 5).

To verify that the smaller CUVAF area in the myopic group of both medical and environmental science cohorts was not due to the protective effect of spectacles, a new comparison was performed analysing only subjects that wear spectacles, in both the myopic and control groups (individuals wearing spectacles for hyperopia or astigmatism), and it was found that the CUVAF areas in myopic and HM were still significantly smaller than in controls (*p* < 0.05) (Figure 6).

### 3.4. Myopia Hereditary Factors and CUVAF Genetic Factors

When analysing the entire study population, the majority of participants in the control group had no myopic parents (52.0%) and very low percentages (37.3% and 10.7%) had one or both parents with myopia, respectively, whereas significantly low percentages of participants having no myopic parents (*p* < 0.01 in both cases) were found in M2 and HM groups, and a significantly higher percentage having one or both myopic parents (*p* < 0.01 and *p* < 0.001, respectively) (Table 3). In addition, having both myopic parents was found to confer 6.2 times more risk of developing early-onset myopia compared to having no myopic parents (*p* = 0.004; OR = 6.2 (95% CI 1.8–20.5)) (Table 3).

Within the group of myopic subjects of both environmental science and medical students, the age of myopia onset was significantly lower in the groups with higher degrees of myopia compared to the less myopic group (*p* < 0.01 and *p* < 0.001, respectively), and early-onset myopic subjects had 6.3 times more risk of developing HM than late-onset myopic subjects (*p* = 0.003; OR = 6.3 (95% CI 1.7–23.1)) (Table 1 and Table 2).

In the medical students cohort, a significantly higher number (*p* < 0.05) of subjects belonging to the M2 and HM groups reported a subjective increase in myopia degree since starting medical school compared to those belonging to the M1 group (Table 1). These differences were non-significant in the environmental science students cohort (Table 2).

The allele frequencies of the SNP rs1060043 showed a frequency of 96% for the G allele with no significant differences being found between the different groups studied in both cohorts, as well as with respect to the CUVAF areas (Table 3).

## 4. Discussion

Myopia is the most common refractive error worldwide and has doubled since the last century [1]. This cannot be explained by genetic factors alone; therefore, environmental factors such as less time spent outdoors may play an important role. Hence, the main objective of this cross-sectional study was to analyse whether outdoor exposure could exert a protective effect against the development of myopia and HM in a cohort of young adults, the population that experiences the highest increase in refractive error [33,34], and to investigate CUVAF as an objective biomarker of time spent outdoors. Using a homogeneous cohort of medical and environmental science students, in terms of its geography, age, socioeconomic status, education, and near work activities, with time spent outdoors being the only differentiator, we eliminated potential confounding factors, which allowed us to adequately study the effect of the relationship between time spent outdoors on myopia and HM prevalence, and CUVAF area. This cohort was validated with data provided by the Universidad de Navarra and was representative of the medical and environmental science student population.

After analysing both cohorts of students, it was found that the percentage of myopic individuals was significantly lower in the cohort of environmental science students (50%) in comparison with the medical student cohort (66.66%). In fact, it was found that being a medical student conferred more than double the risk of being myopic in comparison to being an environmental science student. Moreover, in the medical students cohort, there was a significantly greater number of subjects in the moderate myopia group (M2). We hypothesise that this could be related with a significatively higher number of medical students reporting an increase in myopia during their university years, so that they might have experienced progression from low to moderate myopia during medical school, while environmental science students were less than half as likely to have moderate myopia than medical students, which means that academic programmes that include several hours of outdoor field work per week could act as a protective factor for myopia progression.

All participants in this study performed near work activities for at least five hours per day, and near work was not significantly associated with myopia, which confirms the homogeneity of the selected cohort. Other studies have likewise found no association between myopia and near work activities in university populations [19], and the SAVES study (The Sydney Adolescent Vascular and Eye Study) revealed that near work may induce early-onset myopia, but only in young children [11]. Conversely, recent research suggests that near work intensity may be more significant than total number of hours [7,8].

As it was previously established, there was a significantly higher percentage of myopic subjects in the medical students cohort in comparison with the environmental science cohort, and the only factor that could differ between them was the hours spent doing outdoor activities. According to data provided by the Universidad de Navarra, the environmental science degree includes several hours of outdoor field work per week as part of the academic programme; in fact, these students showed that they spent significantly more time outdoors than medical students, while their time performing near work activities was similar. This means that outdoor exposure could be acting as a protective factor from the development of myopia in the environmental science cohort; therefore, we concur with findings from previous studies that outdoor activities may contribute to a decrease in myopia prevalence, regardless of near work, which was similar in our two cohorts [11], not only in children but also in adolescents and young adults [11,19,35,36].

In this study, participants with myopia were significantly associated with smaller CUVAF areas than controls in both cohorts of students, especially in the HM group, where greater differences were observed. A statistically significant correlation was obtained between the spherical equivalent and CUVAF areas, where more profound myopia resulted in a smaller CUVAF. This is reaffirmed by the significantly higher number of myopes who had CUVAF areas at the 0th percentile. As CUVAF is a marker for sun exposure, these results could be interpreted to mean that individuals with more profound myopia have smaller CUVAF areas because they spend less time outdoors than non-myopic individuals, and this was confirmed by the questionnaire results and is in line with previous reports [37,38].

Although a correlation between CUVAF area and time spent outdoors reported in the questionnaire was observed in both student cohorts, this correlation did not reach a level of statistical significance. These results are in line with the findings of other authors [28] and may be due to two reasons; first, the data being self-reported and subjective, it is likely that individuals may have over- or underestimated time spent outdoors [39,40]; second, assuming that CUVAF represents lifetime cumulative exposure to UV radiation [23,29,41], the questionnaire only asked about current outdoor activities, meaning that previous sunlight exposure during childhood may also have masked any small changes in CUVAF from recent years.

The analysis of the CUVAF area between medical and environmental science students did not show any significant difference, despite the significant difference in time spent outdoors between both cohorts, meaning that CUVAF represents an objective biomarker of time spent outdoors when comparing control and myopic groups, but it can be affected by device compliance or positioning, skin type, and use of sun protection. In this case, we demonstrated that spectacle-wearing did not modify the CUVAF area size, as previously described in both spectacle and contact lenses wearers [27,28,29], while sunbathing without sun protection may result in larger CUVAF areas without necessarily conferring additional protection against myopia [24], and could act as a confounding factor when measuring CUVAF.

Despite strong evidence linking CUVAF area to sun exposure, the genetic component is reportedly responsible for a 0.37 variation, and the SNP rs1060043 has been associated with CUVAF in the Australian population [32]. However, in this study, no association was found between SNP rs1060043 and CUVAF area; and further studies are needed to determine whether this is due to the non-association of this polymorphism with the Spanish population or due to the small study sample size.

The mean CUVAF area of our study population was 2.93 mm^2^ ± 3.06. This value differs greatly from values observed in studies in Australia, Norfolk Island, Tasmania, and India [23,27,32,42], and is similar to values from Ireland and northern Europe [24,28]. The less intense exposure to UV radiation experienced by European populations could explain these differences. The relationship between CUVAF area and myopia has been studied mainly in the Southern Hemisphere, especially in Australia [26,27], while little research has been conducted in Europe, and only one study has shown an association between CUVAF area and myopia in the Northern Hemisphere [29]. This is the first study to assess the usefulness of CUVAF as a biomarker of outdoor exposure and its inverse association with myopia in a European Mediterranean population; therefore, the current results will be useful for future larger studies at similar latitudes.

Finally, despite the recognised contribution of environmental factors in the increase in myopia prevalence, other factors, such as hereditary factors, also play an important role. In our study, myopia prevalence was associated with the number of myopic parents in a dose-dependent manner. In addition, individuals with two myopic parents had a 6.2 times greater risk of developing early-onset myopia, and subjects from this group were 6.3 times more likely to develop HM. These results support the influence of hereditary factors on the development of myopia, primarily early-onset myopia, which is the most likely to progress to HM, not only because of genetic factors but also because parents and their children may share a common environment.

## 5. Conclusions

In conclusion, our findings confirm the importance of outdoor activities as a protective factor against myopia and HM in young adults, regardless of near work activities. In addition, CUVAF represents an objective, non-invasive biomarker of outdoor exposure that is inversely associated with myopia, although it can be affected by several factors such as the use of sun protection and this should be taken into account. Moreover, in contrast with the Australian study, the SNP rs1060043 did not demonstrate any association with CUVAF in this Spanish cohort. Therefore, although further studies are still needed, this and other studies have shown that CUVAF could be useful as part of the follow-up assessment of myopic children and young adults, who are the most at-risk population.

## Figures and Tables

**Figure 1 jcm-11-04264-f001:**
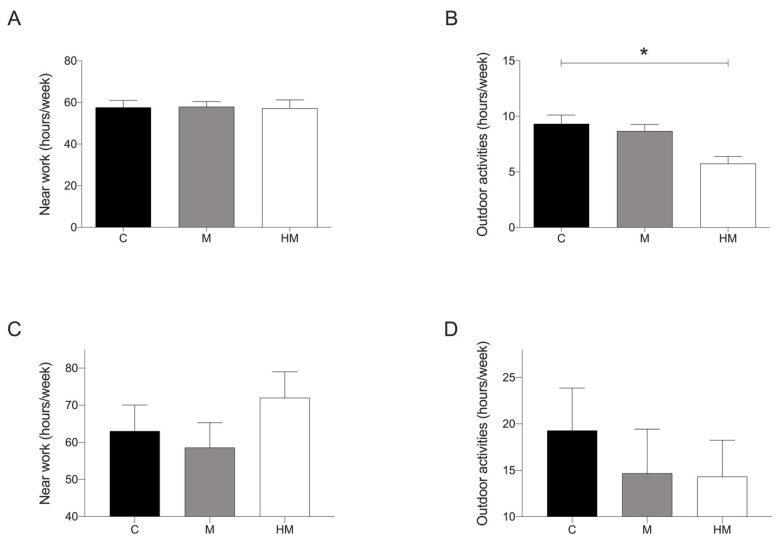
Analysis of environmental factors in both student cohorts. (**A**) Medical students: analysis of weekly hours of near work activities in control group vs. myopic and HM groups. (**B**) Medical students: analysis of weekly hours of outdoor activities in control group vs. myopic and HM groups. (**C**) Environmental science students: analysis of weekly hours of near work activities in control group vs. myopic and HM groups. (**D**) Environmental science students: analysis of weekly hours of outdoor activities in control group vs. myopic and HM groups. C: control group (>1.00 D); M: myopic group (≤1.00 D); HM: High myopia (≤−6.00 D or AL > 26 mm). Significance *p* < 0.05. * *p* < 0.05.

**Figure 2 jcm-11-04264-f002:**
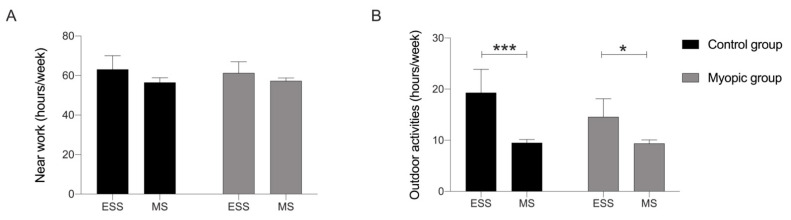
Comparison of environmental factors between the two cohorts of students. (**A**) Weekly hours of near work activities: analysis of control group vs. myopic group between environmental science and medical students. (**B**) Weekly hours of outdoor activities: analysis of control group vs. myopic group between environmental science and medical students. ESS: environmental science students; MS: medical students. Control group (>1.00 D); Myopic group (≤1.00 D). Significance *p* < 0.05. * *p* < 0.05, *** *p* < 0.001.

**Figure 3 jcm-11-04264-f003:**
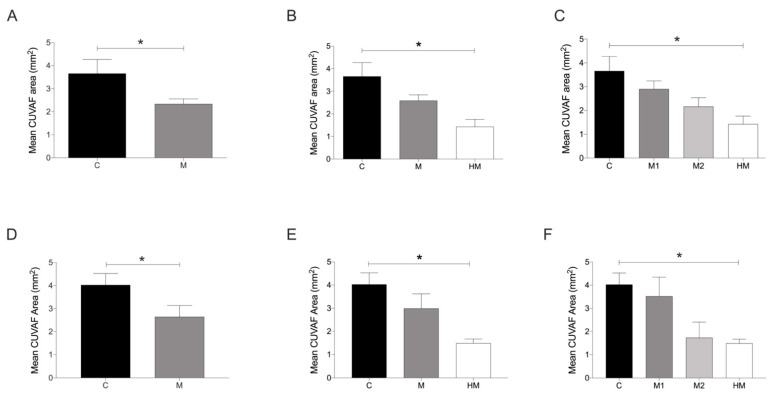
Analysis of differences in mean CUVAF area (mm^2^) between the control and myopic groups in both student cohorts. (**A**) Medical students: control group vs. myopic group. (**B**) Medical students: control group vs. myopic and HM groups. (**C**). Medical students: control group vs. M1, M2, and HM groups. (**D**) Environmental science students: control group vs. myopic group. (**E**) Environmental science students: control group vs. myopic and HM groups. (**F**) Environmental science students: control group vs. M1, M2, and HM groups. CUVAF: conjunctival ultraviolet autofluorescence. C: control group (>1.00 D); M: myopic group (≤1.00 D); M1: −1.00 to −3.00 D; M2: −3.25 to −5.75 D; HM: High myopia (≤−6.00 D or AL > 26 mm). Significance *p* < 0.05. * *p* < 0.05.

**Figure 4 jcm-11-04264-f004:**
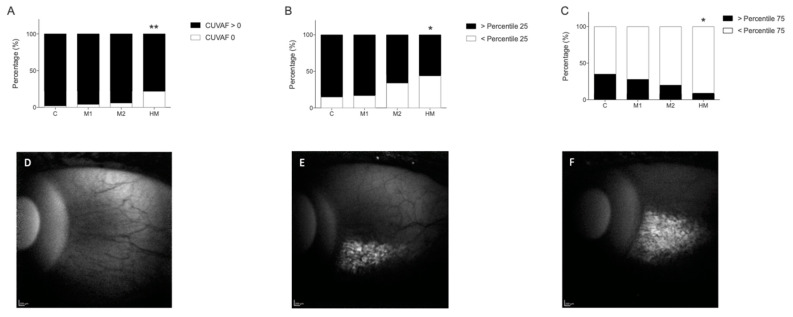
**Top row.** Frequency of individuals in the study population with CUVAF areas at the 0th, 25th, and 75th percentiles in groups C, M1, M2, and HM. (**A**) Percentage of individuals with CUVAF (mm^2^) at the 0th percentile. (**B**) Percentage of individuals with CUVAF (mm^2^) at the 25th percentile. (**C**) Percentage of individuals with CUVAF (mm^2^) at the 75th percentile. **Bottom row.** Example of CUVAF areas of individuals belonging to the different percentiles. (**D**) Example of CUVAF area of the 0th percentile. (**E**) Example of CUVAF area of the 25th percentile. (**F**) Example of CUVAF area of the 75th percentile. CUVAF: conjunctival ultraviolet autofluorescence. C: control group (>1.00 D); M: myopic group (≤1.00 D); M1: −1.00 to −3.00 D; M2: −3.25 to −5.75 D; HM: High myopia (≤−6.00 D or AL > 26 mm). Significance *p* < 0.05. * *p* < 0.05, ** *p* < 0.01.

**Figure 5 jcm-11-04264-f005:**
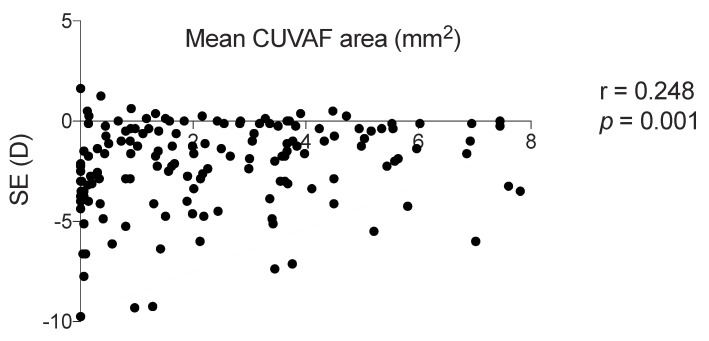
Pearson correlation analysis between spherical equivalent and mean CUVAF area (mm^2^). CUVAF: conjunctival ultraviolet autofluorescence. D: dioptres. SE: spherical equivalent. Significance *p* < 0.05.

**Figure 6 jcm-11-04264-f006:**
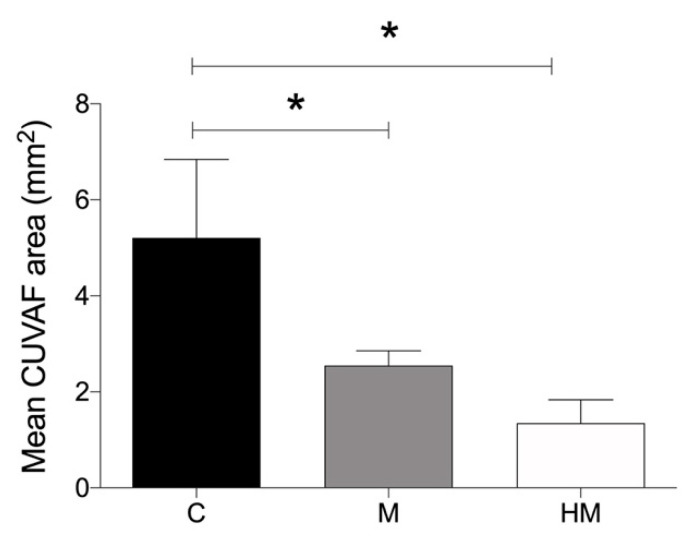
Analysis of mean CUVAF area (mm^2^) in spectacle wearers of the control and myopic groups. CUVAF: conjunctival ultraviolet autofluorescence. C: control group (>1.00 D); M: myopic group (≤1.00 D); M1: −1.00 to −3.00 D; M2: −3.25 to −5.75 D; HM: High myopia (≤−6.00 D or >26 mm LA). Significance *p* < 0.05. * *p* < 0.05.

**Table 1 jcm-11-04264-t001:** Demographic and ophthalmological characteristics of the medical students.

	Total	Control Group	M1	M2	HM	*p*-Value
Number of Participants (%)	156 (100%)	52 (33.33%)	46 (29.49%)	35 (22.44%)	23 (14.74%)	-
Age (Mean ± SD)	22.38 ± 0.85	22.35 ± 0.93	22.54 ± 0.81	22.23 ± 0.77	22.39 ± 0.89	0.410
Female Gender (%)	109 (69.87%)	33 (63.46%)	34 (73.91%)	28 (80%)	14 (60.86%)	0.280
Spherical Equivalent (Dioptres ± SD)	−2.18 ± 2.30	−0.04 ± 0.60	−1.62 ± 0.48 ***	−3.60 ± 0.81 ***	−5.98 ± 2.24 ***	**<0.0001**
Axial Length (mm ± SD)	24.43 ± 1.24	23.51 ± 0.82	24.10 ± 0.77 ***	24.93 ± 0.67 ***	26.41 ± 0.74 **	**<0.0001**
Near work (hours/week ± SD)	57.61 ± 22.19	57.65 ± 24.26	57.63 ± 25.30	57.74 ± 16.77	57.26 ± 19.15	0.990
Outdoor activities (hours/week ± SD)	9.03 ± 6.40	9.33 ± 5.67	8.20 ± 5.92	9.27 ± 5.40	5.76 ± 3.04 *	0.035
Mean CUVAF area (mm^2^ ± SD)	2.86 ± 3.23	3.66 ± 4.41	2.90 ± 2.33	2.40 ± 2.58	1.67 ± 1.90 *	0.023
Age of myopia onset (Mean ± SD)	13.80 ± 4.46	-	19 ± 3.12	12.41 ± 2.44 ***	10.04 ± 4.4 ***	**<0.001**
Increased myopia during university years (%)	88 (84.61%)	-	34 (73.91%)	33 (94.29%) *	21 (91.30%) *	**0.018**

HM, high myopia; SD, standard deviation; CUVAF, conjunctival ultraviolet autofluorescence, Control group: >1.00 D; M1: −1.00 to −3.00 D; M2: −3.25 to −5.75 D; HM: ≤−6.00 D or AL > 26 mm. Significance *p* < 0.05. * *p* < 0.05, ** *p* < 0.01, *** *p* < 0.001.

**Table 2 jcm-11-04264-t002:** Demographic and ophthalmological characteristics of environmental science students.

	Total	Control Group	M1	M2	HM	*p*-Value
Number of Participants (%)	52 (100%)	26 (50%)	14 (26.92%)	6 (11.54%)	6 (11.54%)	-
Age (Mean ± SD)	22.36 ± 1.73	22.66 ± 1.97	22.1 ± 1.6	20.66 ± 0.58	22 ± 1	0.34
Female Gender (%)	36 (69.23%)	18 (69.23%)	10 (71.43%)	6 (100%)	2 (33.33%)	0.28
Spherical Equivalent (Dioptres ± SD)	−1.64 ± 1.54	−0.45 ± 0.56	−1.82 ± 0.65 ****	−3.71 ± 0.69 ****	−4.29 ± 0.07 ****	**<0.0001**
Axial Length (mm ± SD)	23.15 ± 1.28	23.42 ± 1.00	24.08 ± 0.22	25.30 ± 0.23 **	26.33 ± 0.4 ****	**<0.0001**
Near work (hours/week ± SD)	55.30 ± 21.83	63.07 ± 25.16	56.8 ± 7.56	61.67 ± 33.29	72 ± 9.90	0.88
Outdoor activities (hours/week ± SD)	17.15 ± 14.34	19.31 ± 16.43	16.2 ± 15.64	12.17 ± 11.34	14.33 ± 6.81	0.87
Mean CUVAF area (mm^2^ ± SD)	3.33 ±2.64	4.03 ±2.59	3.52 ± 3.21	1.73 ± 1.82	1.49 ± 0.50	0.34
Age of myopia onset (Mean ± SD)	12.69 ± 5.62	-	16.7± 3.67	7 ± 4.36 **	8.7 ± 1.52 **	**0.0036**
Increased myopia during university years (%)	16 (30.81)	-	8 (57.10%)	4 (66.66%)	4 (66.66%)	0.18

HM, high myopia; SD, standard deviation; CUVAF, conjunctival ultraviolet autofluorescence, Control group: >1.00 D; M1: −1.00 to −3.00 D; M2: −3.25 to −5.75 D; HM: ≤−6.00 D or AL > 26 mm. Significance *p* < 0.05. ** *p* < 0.01, **** *p* < 0.0001.

**Table 3 jcm-11-04264-t003:** Myopia hereditary factors and CUVAF genetic factors in the entire student population. Calculation of frequencies with respect to the control group using Fisher’s F test.

	Total	Control Group	M1	M2	HM
No myopic parents (%)	72 (35.3%)	39 (52.0%)	22 (37.3%)	7 (17.0%) **	4 (13.8%) **
One myopic parent (%)	85 (41.7%)	28 (37.3%)	24 (40.7%)	24 (58.6%) **	9 (31.0%)
Both myopic parents (%)	47 (23.0%)	8 (10.7%)	13 (22.0%) *	10 (24.4%) **	16 (55.2%) ***
Rs1060043 Allele G/A (%)	300/12 (96/4)	101/3 (97/3)	87/5 (95/5) ns	67/3 (96/4) ns	45/1 (98/2) ns

CUVAF: conjunctival ultraviolet autofluorescence. HM: High myopia. ns: non-significant. Control group: >1.00 D; M1: −1.00 to −3.00 D; M2: −3.25 to −5.75 D; HM: ≤−6.00 D or AL > 26 mm. Significance *p* < 0.05. * *p* < 0.05, ** *p* < 0.01, *** *p* < 0.001.

## Data Availability

Not applicable.

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
