# Peer review of "A Cross-Sectional Observational Study of the Relationship between Outdoor Exposure and Myopia in University Students, Measured by Conjunctival Ultraviolet Autofluorescence (CUVAF)"

_jcm, 2022, doi:10.3390/jcm11154264_

Round 1
Reviewer 1 Report
The authors designed a study to assess the level of outdoor activity and leve of myopia in two groups, medical students and environmental study students. They found that medical students had more myopia and less outdoor activity. This was also measured by an objective assessment of sun exposure using conjunctival autofluorescence.
The study is well done with sound methods and analysis, and well written.
I would only recommend that in the discussion please acknowledge that this is not a randomized clinical trial, it is a cross sectional study. Therefore, the study is susceptible to confounding and potential spurious associations. Having said that, this study contributes to the building evidence that outdoor light exposure is protective for myopia.
Reviewer 2 Report
This paper is interesting. Meanwhile, there are several concerns.
1. How long dose CUVAF quantify the amount of time spent outdoors? Is it a value for a certain period of time, like hemoglobin A1c? For example, could CUVAF be affected by whether an individual has been outdoors one day prior?
2. Furthermore, could CUVAF values be affected by the use of glasses or contact lenses, even if they were used during outdoor activities? What was the association between their use and CUVAF values in this study?
3. In abstract section, abbreviation of CUVAF should be defined at first mention (Line 27) of the term prior its use to denote the corresponding term in all subsequent mentions.
4. In statistical analyses section, have the authors analysed the normality? Please describe.
5. In tables 1 and 2, there is a typographical error in term of spherical equivalent.
6. In tables 1 and 2, how did the authors investigate the increased myopia during university years? Please describe as it was difficult to understand.
7. In discussion section, please explain why the authors found a significant difference between the control group and HM in B of figure 1, whereas they did not show a significant difference between the control group and HM in D.
Round 2
Reviewer 2 Report
I thank the authors for their adequately responding to my comments.